psychology

locus-of-control, smoking, tobacco, drinking, alcohol, ALSPAC

**Author for correspondence:**
G. Lassi
e-mail: glenda.lassi@astrazeneca.com

# Locus of control is associated with tobacco and alcohol consumption in young adults of the Avon Longitudinal Study of Parents and Children

G. Lassi[1,2,3], A. E. Taylor[4,5], L. Mahedy[4], J. Heron[4], T. Eisen[6,7] and M. R. Munafò[1,3]

[1]MRC Integrative Epidemiology Unit (IEU) at the University of Bristol, Bristol, UK
[2]Centre for Genomics Research, Precision Medicine and Genomics, IMED Biotech Unit, AstraZeneca, Cambridge, UK
[3]UK Centre for Tobacco and Alcohol Studies, School of Experimental Psychology, University of Bristol, Bristol, UK
[4]Population Health Sciences, Bristol Medical School, University of Bristol, Bristol, UK
[5]NIHR Biomedical Research Centre at the University Hospitals Bristol NHS Foundation Trust and the University of Bristol, Bristol, UK
[6]Oncology Translational Medicine Unit, Early Clinical Development, IMED Biotech Unit, AstraZeneca, Cambridge, UK
[7]Department of Oncology, University of Cambridge, Cambridge Biomedical Campus, Cambridge, UK

GL, 0000-0001-5102-3294

Individuals appraise events as a consequence of their own actions (i.e. internal locus of control, LoC) or as the outcome of chance or others' will (i.e. external LoC). We hypothesized that having a more external LoC would be associated with higher risk of tobacco and alcohol use. Few studies have examined this association using large prospective data. We evaluated within the Avon Longitudinal Study of Parents and Children (ALSPAC) the associations between LoC at 16 and tobacco and alcohol consumption at 17 and 21 years using logistic regression. A more external LoC at age 16 ($N = 4656$) was associated with higher odds of being a weekly smoker at age 17 (OR 1.18, 95% CI 1.10–1.25) and 21 (OR 1.14, 95% CI 1.07–1.21) and with dependence measured using the Fagerström Test of Nicotine Dependence at age 17 (OR 1.26, 95% CI 1.05–1.51) and 21 (OR 1.25, 95% CI 1.05–1.49). Individuals with external LoC at age 16 were more likely to be hazardous drinkers according to the Alcohol Use Disorders Identification Test at age 17 (OR 1.09, 95% CI 1.04–1.15) but not at 21 (OR 1.01, 95% CI 0.96–1.06). Having a more external LoC at age 16 is associated with

increased tobacco consumption at age 17 and 21 and alcohol consumption at 17 years. LoC may represent an intervention target for preventing substance use and dependence.

# 1. Introduction

Adolescence and young adulthood are critical risk periods for the initiation of tobacco and alcohol use [1]. Studies on brain development support the idea that the maturing brain may be particularly vulnerable to the effects of drugs [2], and that drug use during early life may increase a young person's risk for developing a substance use disorder later in life [3–5]. Tobacco and alcohol consumption are two of the most important preventable causes of morbidity and early mortality. In the UK 8% of 15 year olds smoke tobacco [6] and smoking causes around 79 000 preventable deaths in England costing our economy in excess of £11 billion per year [7]. Moreover, between 2015 and 2016, over 330 000 individuals aged 16 or older were admitted to a hospital as a consequence of alcohol misuse [8].

Understanding factors that influence the use and misuse of tobacco and alcohol is important to inform prevention efforts. In addition to societal factors (e.g. drug availability), psychological characteristics are important in determining whether an individual will go on to use substances, and their subsequent degree of dependence if they do [9–11]. Where these characteristics are modifiable, they offer potential targets for prevention of substance use. One such trait is locus of control (LoC), namely one's perception of control over life events. Individuals differ in the extent to which they judge that events in their lives are a consequence of their own actions or the result of external factors. Those who believe that events are largely the result of their own actions have an internal LoC, whereas those who believe that events are the result of chance or the actions of others have an external LoC. This is thought to arise from associations between behaviour and reinforcers [12], which are first experienced in early life with the primary attachment figure [13]. Interestingly, LoC changes across the lifespan; it tends to become more internal in adolescence, when parental and societal rules are internalized and children's thinking becomes more abstract, hypothetical and critical. It then remains relatively stable in adolescence [14] and adulthood [15] with a trend to becoming yet more internal in later adulthood [16].

Although LoC has been extensively investigated in the psychological and social science literature, there has been little research in the field of substance abuse. Some studies of tobacco and alcohol users have assessed constructs related to LoC such as self-efficacy, for example, by appraising confidence in being able to successfully stop smoking, abstain from alcohol, or resist peer pressure [17–19]. Others have highlighted the role of personality traits as risk factors for drug use and abuse, such as low self-esteem, high sensation seeking [20,21], self-control and, in particular, the inability to inhibit impulsive actions or delay gratification; all are positively associated with addictive behaviour [22]. In addition, internalizing (depression and anxiety) and externalizing (conduct problems) symptoms have also been found to be positively associated with smoking [23,24] and alcohol consumption [25], respectively.

Despite the evidence that LoC is associated with the use/abuse of substances [26], and in contrast with more widely studied behaviours such as cessation and abstinence in adults, data about the relationship between LoC orientation and tobacco and/or alcohol use in young adults is limited. The studies that have examined this have reported that adolescent smokers with a higher nicotine dependence and individuals with greater alcohol consumption, had a more external LoC; however, all of these studies have been limited by relatively small sample sizes and/or by using cross-sectional designs that do not allow to analyse behaviour over a period of time [27–34].

Two large studies have assessed the longitudinal relationship between LoC (or closely related constructs) and smoking. In the British 1946 Birth Cohort, adolescent self-organization was negatively associated with the number of packs of cigarettes smoked in adulthood. By contrast, there was no clear evidence of an association between self-organization and alcohol consumption across adulthood [35]. LoC scales had not been introduced at the time that the 1946 Birth Cohort was tested, while it was well known when the 1970 British Cohort Study began. In the latter, LoC was assessed at age 10 and it was found that an external LoC was associated with being a smoker at age 30 [36].

In order to assess whether LoC may be a possible target to prevent smoking as well as alcohol use and dependence we used data from a large UK-based prospective birth cohort, the Avon Longitudinal Study of Parents and Children (ALSPAC). Using prospective data provides the opportunity of

investigating measures assessed at different timepoints (e.g. adolescence and early adulthood). The clear temporal ordering of exposure, outcomes and confounders helps to rule out the possibility of reverse causation. This study investigated whether the LoC orientation, considered to be stable at age 16 years [14], was associated with proximal (age 17 years), and distal (age 21 years) assessments of smoking and alcohol consumption. We hypothesized that an external LoC at age 16 years would be associated with greater consumption of tobacco and alcohol at ages 17 and 21 years, respectively.

# 2. Methods

## 2.1. Participants

Participants were selected from the ALSPAC birth cohort (www.alspac.bris.ac.uk). The cohort consists of children born to residents of the former Avon Health Authority area in South West England who had an expected date of delivery between 1 April 1991 and 31 December 1992. The ALSPAC birth cohort consists of 14 541 pregnancies that resulted in 14 062 live births: 13 988 infants were still alive at 1 year [37] and a small number of participants withdrew from the study ($n = 24$). The sample was further restricted to singletons or first-born twins, resulting in a starting sample of 13 775. Detailed information about ALSPAC is available online www.bris.ac.uk/alspac and in the cohort profiles [37,38]. The study website contains details of all the data that is available through a fully searchable data dictionary (http://www.bris.ac.uk/alspac/researchers/data-access/data-dictionary/). Ethics approval for the study was obtained from the ALSPAC Ethics and Law Committee and the Local Research Ethics Committees. We performed a secondary analysis of existing data and no informed consent was needed from participants.

## 2.2. Measures

### 2.2.1. Locus of control

LoC was assessed at 16 years of age (mean 16.7, s.d. 0.2) by means of a brief version of the Nowicki–Strickland Internal-External scale consisting of 12 items [39] (see electronic supplementary material, table S1). Only individuals who answered all items were included. Response options were yes/no with items summed to create a total score (range 0–12). Higher score indicates a more externally oriented LoC.

### 2.2.2. Smoking behaviour

Smoking behaviour was assessed at a research clinic using a computer-assisted survey at age 17 (mean age 17.8 years, s.d. = 0.4), and by a mixture of postal and online questionnaire at 21 years (mean age 20.9 years, s.d. 0.5). Children were asked whether they were smokers and how often they smoked at ages 17 and 21 years. Nicotine dependence (ND) was measured, at the same ages, in daily smokers using the Fagerström Test for Nicotine Dependence (FTND). The total possible score ranged from 0 to 10. As we were interested in examining nicotine dependence, a binary variable (0/1) was created (greater than or equal to 4 indicating moderate to high levels of nicotine dependence) [40].

### 2.2.3. Alcohol consumption

Alcohol misuse was assessed at the same measurement occasions at age 17 and 21 using the Alcohol Use Disorders Identification Test (AUDIT) [41]. This is a 10-item questionnaire where a score less than or equal to 8 reflects non-hazardous drinking, while a score greater than 8 reflects hazardous drinking [42–44].

### 2.2.4. Potential confounders

A range of measures were considered to be potential confounders of the LoC and smoking and drinking behaviours relationships. Confounders were chosen on the basis of previous literature and tested for associations with the exposure and the outcomes. The confounders included were: age, sex, IQ at age 8 (assessed in a focus clinic using a short form of the Wechsler Intelligence Scale for Children, WISC-III), maternal smoking (yes/no information on maternal smoking when the child was 12 years old

were obtained from a self-report postal questionnaire), maternal drinking (mothers completed a postal questionnaire asking about daily alcohol consumption when the child was 12 years old). Reponses, including beverage type and volume consumed, were converted into UK standard units—8 g alcohol. Socio-economic information was also considered and included maternal education (coded as CSE, vocational, O level, A levels and degree) and paternal occupation (coded as professional, managerial and technical, non-manual skilled, manual skilled, partly skilled and unskilled) at child birth.

## 2.3. Statistical analysis

Associations between LoC (measured as a continuous variable) and all outcomes (all binary) were assessed via logistic regressions. All analyses were conducted in Stata (version 14). Models adjusted for all confounders are reported as our main results. Models adjusted for sex and age analyses are reported in electronic supplementary material, tables S6–S7. Results for the complete case samples are reported in the main text.

### 2.3.1. Missing data

Since analysis based on complete cases may be biased [45], we examined possible effects of missing data using multiple imputation. Imputation was performed separately for the age 17 and age 21 outcomes using the 'mi ice' command in Stata. Imputation models contained all exposure, covariate and outcome data as well as other demographic predictors of missing data. We created 50 imputed datasets for each outcome apart from AUDIT at 17 and 21 and FTND at age 21, where we created 75 imputed datasets as inspection of the Monte Carlo error indicated variation was higher than recommended [46]. All the analyses that included imputed data are reported in the electronic supplementary material, tables S8 and S9.

Additional analyses, further adjusting for adverse childhood experiences as well as for conduct disorder, are reported in the electronic supplementary material. In addition to our primary analysis of the associations of LoC at 16 with the three outcomes (smoking status, nicotine dependence and AUDIT), all analyses were also run for LoC assessed at 8 years (see the electronic supplementary material).

# 3. Results

## 3.1. Characteristics of participants

For a detailed description of the study sample, including age, sex, IQ, maternal smoking and drinking, maternal education, paternal social class, smoking status, FTND and AUDIT scores, see electronic supplementary material, table S2.

LoC data were available on 4656 individuals at age 16 years. These individuals tended to have an internal LoC (median = 3, IQR 2,4; mean = 3.21, s.d. 2.12) although there was evidence that LoC was more external in females compared with males. Maternal smoking, when the child was 12 years old, was strongly associated with a more external LoC. A more internal LoC was strongly associated with a higher IQ and socio-economic status (maternal education and paternal occupational) but also with a maternal higher alcohol consumption (electronic supplementary material, table S3).

Furthermore, higher IQ was associated with lower likelihood of being a smoker and lower FTND score; being a smoker was associated with lower maternal education and paternal occupation, maternal smoking as well as with higher maternal alcohol consumption (electronic supplementary material, table S4). Finally, hazardous drinking at age 21 was less common in females but drinking behaviour did not appear to be strongly socially patterned. Maternal alcohol use was associated with alcohol consumption (electronic supplementary material, table S5).

## 3.2. Tobacco Use

### 3.2.1. LoC at 16 and smoking status at 17 and 21 years

There was strong evidence that a more external LoC at age 16 was associated with being at least a weekly smoker at age 17 (OR 1.18, 95% CI 1.10, 1.25, $p < 0.001$) and age 21 (OR 1.14, 95% CI 1.07, 1.21, $p < 0.001$; figure 1 and electronic supplementary material, table S6).

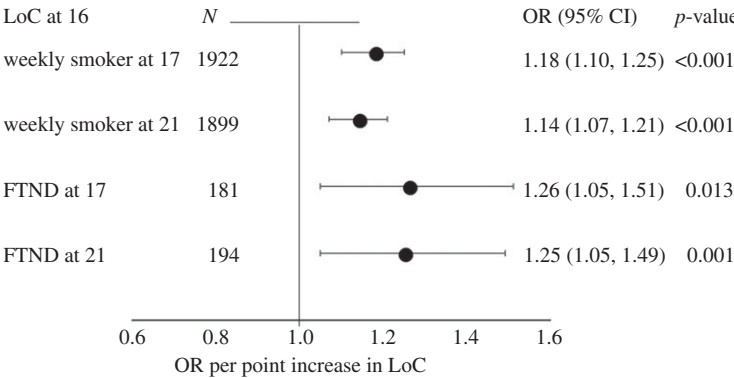

**Figure 1.** Association between locus of control at 16 years and tobacco consumption at 17 and 21 years. LoC, Locus of Control; FTND, Fagerström Test for Nicotine Dependence. Dots represent the odds ratio for being at least weekly smoking compared to less than weekly smoking and for being dependent on nicotine (FTND ≥ 4) compared to not being dependent on nicotine (FTND < 4). Horizontal lines represent 95% CIs. Regressions were adjusted for age, sex, IQ, maternal smoking at 12, maternal drinking at 12, maternal education and paternal occupation.

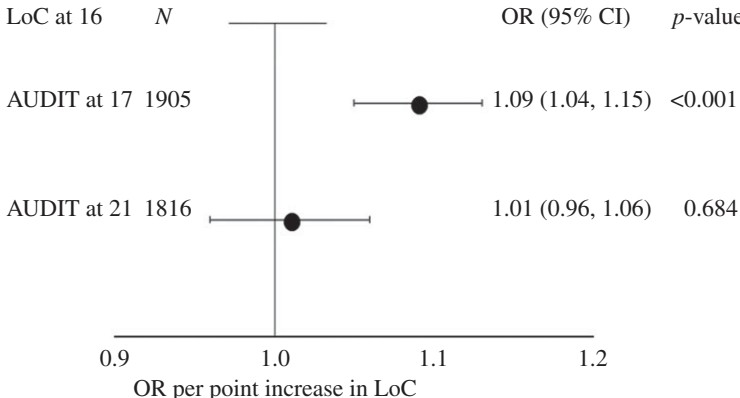

**Figure 2.** Association between locus of control at 16 years and alcohol consumption at 17 and 21 years. LoC, Locus of Control; AUDIT, Alcohol Use Disorders Identification Test. Dots represent the odds ratio for hazardous use of alcohol (AUDIT > 8) compared to non-hazardous (AUDIT ≤ 8) at 17 and 21 years. Horizontal lines represent 95% CIs. Regressions were adjusted for age, sex, IQ, maternal smoking at 12, maternal drinking at 12, maternal education and paternal occupation.

### 3.2.2. LoC at age 16 and nicotine dependence at 17 and 21 years

Having a more external LoC at age 16 was associated with nicotine dependence at age 17 (OR 1.26, 95% CI 1.05–1.51, $p = 0.013$), and age 21 (OR 1.25, 95% CI 1.05–1.49, $p = 0.001$; figure 1 and electronic supplementary material, table S6).

## 3.3. Alcohol use

### 3.3.1. LoC at age 16 and alcohol misuse at 17 and 21 years

A more external LoC at age 16 was associated with higher odds of hazardous drinking on the AUDIT score at age 17 (OR 1.09, 95% CI 1.04, 1.15, $p < 0.001$) but not at age 21 (OR 1.01, 95% CI 0.96–1.06, $p = 0.684$; figure 2 and electronic supplementary material, table S7).

## 3.4. Imputation

Results were similar following multiple imputation of missing exposure and covariate data, although statistical evidence for an association was weaker for LoC at age 16 and FTND at 17 years (electronic supplementary material, tables S8 and S9).

# 4. Discussion

In this study, we found strong evidence that a more external LoC at age 16 was associated with higher odds of being at least a weekly smoker and being nicotine dependent at age 17 and age 21. A more external LoC at age 16 was also strongly associated with greater odds of hazardous alcohol use at age 17 but not at age 21. Importantly, the results remained the same after adjusting for the confounders.

## 4.1. Comparison with previous studies

To the best of our knowledge, this is the largest study to date to assess the prospective association between LoC and tobacco smoking and alcohol use. As hypothesized, we found evidence that perceiving less control over life events at age 16 years, hence having a more external LoC, is associated with proximal (age 17 years) and distal (age 21 years) smoking behaviours. Smoking is known to have negative health outcomes and people that feel more in control over their life may intentionally take action and engage in healthier behaviours [13]. By contrast, those with an external locus of control may be less aware of or less concerned about the consequences of their own behaviour (e.g. the consumption of tobacco) [13].

We also found inconsistent evidence for an association between LoC orientation and alcohol consumption. Although a more external LoC at age 16 was associated with hazardous use of alcohol at 17, there was no such association at 21. Surprisingly, the percentage of hazardous drinkers increased from age 17 ($N = 1029$, 37.26% of the total number of participants that filled the AUDIT at 17 years) to age 21 ($N = 1551$, 54.34% of the total number of participants that filled the AUDIT at 21 years; electronic supplementary material, table S3). This difference may be due to the fact that legal drinking age in the UK is 18 years and at age 21 alcohol is very easily accessible making hazardous drinking common, particularly among certain subgroups (e.g. students) [47]. In order to understand this relationship more fully, exploration of associations between LoC and alcohol consumption at a later age is needed.

When examining the association between LoC and drinking/smoking, previous work has mainly focused on two outcomes: treatment adherence [48] and the ability to remain abstinent [34], which have been both found to be positively associated with having an internal LoC. Interestingly, Segall & Wynd [49] report that individuals who relapse to smoking after having quit have a more external LoC, while Stuart *et al.* [50] found that a successful smoking cessation attempt, and maintenance of abstinence, were both associated with an internal LoC. Similarly, LoC orientation in alcohol-dependent individuals varies if measured over the course of treatment and remaining abstinent during treatment is associated with LoC becoming more internal [34,51,52]. Alcohol use during treatment is associated with external LoC and increased chances of poor treatment outcomes [53]. These findings indicate that LoC can change within the context of treatment, and this plasticity could be used to prevent tobacco and alcohol use and misuse in adolescents.

## 4.2. Limitations

The present study should be considered in light of a number of limitations. First, patterns of confounding of LoC and smoking behaviours are in the same direction (e.g. a more external LoC and weekly smoking are both positively associated with both paternal occupation and maternal education) so it is possible that the observed association between LoC and smoking could be explained by residual confounding. However, residual confounding is unlikely to fully explain this association since LoC at age 8 has similar social patterning but was not associated with smoking behaviours (see the electronic supplementary material). Further studies could investigate whether other variables, such as life stress, may account for a possible mediating pathway between LoC and substance use. Furthermore, alcohol consumption is positively associated with some socio-economic indicators such as family income [54,55] and negatively associated with others such as parents' education level [55,56]. Interestingly, in our analyses, we found that alcohol misuse was positively associated with maternal education but not with paternal occupation (electronic supplementary material, table S5). Statistical adjustment alone may not be adequate to address residual confounding related to social patterning. Second, our study does not allow us to define the causal relationship between LoC and smoking and drinking behaviours, and in turn whether LoC could be the target of preventive interventions to reduce tobacco and alcohol use. One possible way to assess causality is to use animal models and assess

whether experimentally manipulated LoC causes different consumption profiles in laboratory animals exposed to nicotine or alcohol. Another possibility is to use genetic variants as instrumental variables in a Mendelian randomization framework [57] although, at present, we are not aware of any reports of genetic variants associated with LoC. Third, both tobacco and alcohol consumption were self-reported and individuals tend to underestimate their smoking and misreport their drinking [58,59], however, participants completed questionnaires individually and were assured of the anonymity of their responses. Fourth, not all ALSPAC participants completed the questionnaires on smoking and drinking behaviours; therefore, these results may be affected by bias due to failure to follow up. Nevertheless, similar findings were obtained for the complete case and imputed samples.

# 5. Conclusion

In this study, we examined the relationship between the consumption of tobacco and alcohol and LoC orientation in young adults. As LoC is modifiable, if the associations observed here are causal, LoC may represent a potential target for intervention. Our results suggest that if this is the case such interventions may need to be delivered during certain critical periods, as only LoC at age 16 (and not at age 8) was associated with tobacco and alcohol use. In addition, adolescence is a key period for targeting prevention strategies for smoking uptake and alcohol misuse. Such interventions could be delivered in the form of teacher-delivered interactive sessions to raise awareness of the effect of psychological factors on smoking and drinking in secondary schools. Successful randomized controlled trials targeting maladaptive coping strategies and personality traits, including hopelessness, sensation seeking, anxiety and sensitivity, have been conducted in school settings in order to prevent alcohol and drug use (with no specific reference to tobacco consumption though) in high-risk adolescents [60–63].

In addition, since LoC is the perception of one's control over life events, it is reasonable to consider that such perception can be influenced by biased information processing, that is, by cognitive biases [64]. By targeting eventual distorted selectivity in perceiving and elaborating information of one's experiences and preferences, LoC orientation could be steered towards a perception of control.

An intervention aimed at enhancing an internal LoC could develop ownership of actions by experiencing the direct association between behaviour and its effects and favour a sense of responsibility and control over one's actions.

Ethics. Ethics approval for the study was obtained from the ALSPAC Ethics and Law Committee and the Local Research Ethics Committees.

Data accessibility. Data used for this submission will be made available on request to the Executive (alspac-exec@bristol.ac.uk). The ALSPAC data management plan (http://www.bristol.ac.uk/alspac/researchers/data-access/documents/alspac-data-management-plan.pdf) describes in detail the policy regarding data sharing, which is through a system of managed open access.

Authors' contributions. G.L. conceived the study, analysed and interpreted the data and wrote the manuscript. A.E.T. analysed and interpreted the data; L.M. and J.H. analysed the data; M.R.M. interpreted the data. All authors reviewed and revised multiple drafts of the paper and gave final approval for publication.

Competing interests. The authors have no conflict of interests to declare.

Funding. This work was supported by the Medical Research Council and the University of Bristol (MC_UU_12013/6). G.L. and M.R.M. are members of the UK Centre for Tobacco and Alcohol Studies, a UKCRC Public Health Research: Centre of Excellence. G.L. was supported by an AstraZeneca postdoctoral fellowship. J.H. and L.M. were supported by the MRC and Alcohol Research UK (MR/L022206/1). We also acknowledge support from The Centre for the Development and Evaluation of Complex Interventions for Public Health Improvement (DECIPHer), a UKCRC Public Health Research Centre of Excellence (joint funding (MR/KO232331/1) from the British Heart Foundation, Cancer Research UK, Economic and Social Research Council, Medical Research Council, the Welsh Government and the Wellcome Trust (under the auspices of the UK Clinical Research Collaboration) and the NIHR School of Public Health Research. M.R.M. and L.M. are supported by the NIHR Biomedical Research Centre at the University Hospitals Bristol NHS Foundation Trust and the University of Bristol. The views expressed in this publication are those of the authors and not necessarily those of the NHS, the National Institute for Health Research or the Department of Health and Social Care.

Acknowledgements. We are extremely grateful to all the families who took part in this study, the midwives for their help in recruiting them, and the whole ALSPAC team, which includes interviewers, computer and laboratory technicians, clerical workers, research scientists, volunteers, managers, receptionists and nurses. The UK Medical Research Council and Wellcome Trust (Grant ref: 102215/2/13/2) and the University of Bristol provide core support for ALSPAC.

Disclaimer. This publication is the work of the authors and Glenda Lassi, Amy Taylor, Liam Mahedy, Jon Heron, Tim Eisen and Marcus Munafò will serve as guarantors for the contents of this paper.

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
