## [Reviewer comments · Royal Society Open Science]

Review History

RSOS-181133.R0 (Original submission)

Review form: Reviewer 1

Is the manuscript scientifically sound in its present form?

Yes

Are the interpretations and conclusions justified by the results?

Yes

Is the language acceptable?

Yes

Is it clear how to access all supporting data?

Yes

Do you have any ethical concerns with this paper?

No

Have you any concerns about statistical analyses in this paper?

Yes

Recommendation?

Accept with minor revision (please list in comments)

Comments to the Author(s)

I found this a clear and useful paper, relating locus of control to smoking and alcohol use. I have a few comments that need clarification and some minor suggestions.

Main comments:

- I think the rationale for using this prospective design could be made more clear (at P5 L12).

Why is specifically this link of interest?

- Why was it specifically necessary to dichotomize the left-skewed FTND? This seems quite extreme. Were the results strongly dependent on this analysis step?

- I'm afraid I didn't understand how age could be used as a confounder (P6 L 44), given that age is already built in to the prospective design?

- Was there no more recent measure of IQ or executive functioning than the WISC-III at age 8?

Minor comments:

- P4, L4: The first comma seems unnecessary, and a "dearth" seems a little overstated given the following paragraph.

- P4, L36: Please explain what the relevant limitation of a cross-sectional design is.

- P7, L16: Capitalization of "model".

- Please provide a reference in the sentence on external locus of control, P10 L 4.

- Final sentence: I think it would be more correct to say "...could develop..."

- The idea of interventions is interesting but not very developed. Maybe more out of possible interest to the authors, they might consider looking at the Cognitive Bias Modification literature.

Review form: Reviewer 2 (Joseph Boden)**Is the manuscript scientifically sound in its present form?**

Yes

Are the interpretations and conclusions justified by the results?

Yes

Is the language acceptable?

Yes

Is it clear how to access all supporting data?

Yes

Do you have any ethical concerns with this paper?

No

Have you any concerns about statistical analyses in this paper?

No

Recommendation?

Accept with minor revision (please list in comments)

Comments to the Author(s)

The present paper uses data from the ALSPAC study to examine linkages between locus of control in adolescence, and subsequent tobacco and alcohol use. The results suggest that there were relatively weak but discernable associations between an externally-focussed locus of control at age 16, and subsequent smoking, and a somewhat less robust association between an external locus of control and hazardous drinking (significant at age 17 but not at age 21).

This is an interesting paper that investigates the possible role of a very old construct in psychology in an epidemiological format. I have the following general comments to make:

1. Because the associations are relatively weak (increase in odds for smoking were 14% to 18%, and for hazardous drinking 9% and 1%), these may be particularly vulnerable to unmeasured confounding. However, it's not at all clear what unmeasured factors could confound these associations. From our own longitudinal research, three possibilities come to mind: childhood behaviour problems (conduct problems; attention problems), exposure to abuse/neglect, and family instability. I don't know whether any of these are significantly associated with the predictor (locus of control), but it seems reasonable to expect that they might be. Do the authors have any data concerning these factors? Are they associated with LoC? If so, it would be helpful to consider these as possible confounders.
2. The authors mention that it is not clear what mechanism links LoC to substance use behaviour. One possibility that springs to mind is life stress. It might be possible, for example, to set up a mediation model in which LoC at 16 predicts life stress (if measured, of course) at 17 or 21, to account for a possible mediating pathway. If the authors had such data available, it would add value to this analysis.

Decision letter (RSOS-181133.R0)

17-Dec-2018

Dear Dr Lassi,

The editors assigned to your paper ("Locus of control is associated with tobacco and alcohol consumption in young adults of the Avon Longitudinal Study of Parents and Children") have now received comments from reviewers. We would like you to revise your paper in accordance with the referee and Associate Editor suggestions which can be found below (not including confidential reports to the Editor). Please note this decision does not guarantee eventual acceptance.

Please submit a copy of your revised paper before 09-Jan-2019. Please note that the revision deadline will expire at 00.00am on this date. If we do not hear from you within this time then it will be assumed that the paper has been withdrawn. In exceptional circumstances, extensions may be possible if agreed with the Editorial Office in advance. We do not allow multiple rounds of revision so we urge you to make every effort to fully address all of the comments at this stage. If deemed necessary by the Editors, your manuscript will be sent back to one or more of the original reviewers for assessment. If the original reviewers are not available, we may invite new reviewers.

- Data accessibility

If you wish to submit your supporting data or code to Dryad (<http://datadryad.org/>), or modify your current submission to dryad, please use the following link:
<http://datadryad.org/submit?journalID=RSOS&manu=RSOS-181133>

- Competing interests

- Authors' contributions

- Acknowledgements

- Funding statement

on behalf of Dr Joshua Buckholtz (Associate Editor) and Antonia Hamilton (Subject Editor)
openscience@royalsociety.org

Comments to Author:

Reviewers' Comments to Author:

Reviewer: 1

Comments to the Author(s)

I found this a clear and useful paper, relating locus of control to smoking and alcohol use. I have a few comments that need clarification and some minor suggestions.

Main comments:

- I think the rationale for using this prospective design could be made more clear (at P5 L12).

Why is specifically this link of interest?

- Why was it specifically necessary to dichotomize the left-skewed FTND? This seems quite extreme. Were the results strongly dependent on this analysis step?

- I'm afraid I didn't understand how age could be used as a confounder (P6 L 44), given that age is already built in to the prospective design?

- Was there no more recent measure of IQ or executive functioning than the WISC-III at age 8?

Minor comments:

- P4, L4: The first comma seems unnecessary, and a "dearth" seems a little overstated given the following paragraph.

- P4, L36: Please explain what the relevant limitation of a cross-sectional design is.

- P7, L16: Capitalization of "model".

- Please provide a reference in the sentence on external locus of control, P10 L 4.

- Final sentence: I think it would be more correct to say "...could develop..."

- The idea of interventions is interesting but not very developed. Maybe more out of possible interest to the authors, they might consider looking at the Cognitive Bias Modification literature.

Reviewer: 2

Comments to the Author(s)

The present paper uses data from the ALSPAC study to examine linkages between locus of control in adolescence, and subsequent tobacco and alcohol use. The results suggest that there were relatively weak but discernable associations between an externally-focussed locus of control at age 16, and subsequent smoking, and a somewhat less robust association between an external locus of control and hazardous drinking (significant at age 17 but not at age 21).

This is an interesting paper that investigates the possible role of a very old construct in psychology in an epidemiological format. I have the following general comments to make:

1. Because the associations are relatively weak (increase in odds for smoking were 14% to 18%, and for hazardous drinking 9% and 1%), these may be particularly vulnerable to unmeasured confounding. However, it's not at all clear what unmeasured factors could confound these associations. From our own longitudinal research, three possibilities come to mind: childhood behaviour problems (conduct problems; attention problems), exposure to abuse/neglect, and family instability. I don't know whether any of these are significantly associated with the predictor (locus of control), but it seems reasonable to expect that they might be. Do the authors have any data concerning these factors? Are they associated with LoC? If so, it would be helpful to consider these as possible confounders.
2. The authors mention that it is not clear what mechanism links LoC to substance use behaviour. One possibility that springs to mind is life stress. It might be possible, for example, to set up a mediation model in which LoC at 16 predicts life stress (if measured, of course) at 17 or 21, to account for a possible mediating pathway. If the authors had such data available, it would add value to this analysis.

Author's Response to Decision Letter for (RSOS-181133.R0)

See Appendix A.

Decision letter (RSOS-181133.R1)

20-Feb-2019

Dear Dr Lassi:

On behalf of the Editors, I am pleased to inform you that your Manuscript RSOS-181133.R1 entitled "Locus of control is associated with tobacco and alcohol consumption in young adults of the Avon Longitudinal Study of Parents and Children" has been accepted for publication in Royal Society Open Science subject to minor revision in accordance with the referee suggestions. Please find the referees' comments at the end of this email.

The reviewers and Subject Editor have recommended publication, but also suggest some minor revisions to your manuscript. Therefore, I invite you to respond to the comments and revise your manuscript.

- Ethics statement

- Data accessibility

<http://datadryad.org/submit?journalID=RSOS&manu=RSOS-181133.R1>

- Competing interests

- Authors' contributions

- Acknowledgements

- Funding statement

Because the schedule for publication is very tight, it is a condition of publication that you submit the revised version of your manuscript before 01-Mar-2019. Please note that the revision deadline will expire at 00.00am on this date. If you do not think you will be able to meet this date please let me know immediately.

Kind regards,
Andrew Dunn

Royal Society Open Science Editorial Office
Royal Society Open Science
openscience@royalsociety.org

on behalf of Dr Joshua Buckholtz (Associate Editor) and Antonia Hamilton (Subject Editor)
openscience@royalsociety.org

Associate Editor Comments to Author (Dr Joshua Buckholtz):

The authors should be commended for a revision that I consider entirely responsive to the issues raised in the first round of review. My only additional suggestion is that you include the analyses suggested by R2 (i.e. the ACE confound analyses) in supplementary material.

Author's Response to Decision Letter for (RSOS-181133.R1)

See Appendix B.

Decision letter (RSOS-181133.R2)

22-Feb-2019

Dear Dr Lassi,

I am pleased to inform you that your manuscript entitled "Locus of control is associated with tobacco and alcohol consumption in young adults of the Avon Longitudinal Study of Parents and Children" is now accepted for publication in Royal Society Open Science.

on behalf of Dr Joshua Buckholtz (Associate Editor) and Professor Antonia Hamilton (Subject Editor)

Follow Royal Society Publishing on Twitter: [@RSocPublishing](https://twitter.com/RSocPublishing)

Appendix A

Dear Editor,

We thank the reviewers for their comments and the thorough analysis of the manuscript. Please find below a list of all the changes made.

REVIEWERS' COMMENTS:

Reviewer #1

- I think the rationale for using this prospective design could be made more clear (at P5 L12). Why is specifically this link of interest?

We have added a sentence 'The clear temporal ordering of exposure, outcomes and confounders helps to rule out the possibility of reverse causation' and rephrased the following paragraph to stress the importance of using a stable LoC measure that preceded the smoking and alcohol consumption measures. In addition, using data from a prospective design allowed us to test the association between LoC and smoking and alcohol consumption both at age 17 years, and later on, at 21 years.

- Why was it specifically necessary to dichotomize the left-skewed FTND? This seems quite extreme. Were the results strongly dependent on this analysis step?

We were interested in examining non-trivial levels of substance use, therefore we used a binary (hazardous vs non-hazardous) outcome for alcohol consumption too. To this end, in both cases the cutoff points used in this study are commonly used. We have changed the wording in the text to avoid confusion, as follows:

'Nicotine dependence (ND) was measured at the same ages in daily smokers using the Fagerström Test for Nicotine Dependence (FTND). The total possible score ranged from 0 to 10. As we were interested in examining nicotine dependence, a binary variable (0/1) was created (≥ 4) indicating moderate to high levels of nicotine dependence (40).'

- I'm afraid I didn't understand how age could be used as a confounder (P6 L 44), given that age is already built in to the prospective design?

We included age because clinics were run across several months. For example, the mean age of the ALSPAC participants for the 17-years-clinics was 17.8 years, SD = 0.4.

- Was there no more recent measure of IQ or executive functioning than the WISC-III at age 8?

We used IQ at 8 years to ensure that this measure could not be affected by smoking or drinking, thus helping to rule out the possibility of reverse causation.

- P4, L4: The first comma seems unnecessary, and a "dearth" seems a little overstated given the following paragraph.

We have changed '...dearth of research' to 'little research'.

- P4, L36: Please explain what the relevant limitation of a cross-sectional design is.

We have added that cross-sectional studies do not allow to analyse behaviour over a period of time.

- P7, L16: Capitalization of "model".

We have capitalized the 'm'.

- Please provide a reference in the sentence on external locus of control, P10 L 4.

We have added the reference.

- Final sentence: I think it would be more correct to say "...could develop..."

We have amended accordingly.

- The idea of interventions is interesting but not very developed. Maybe more out of possible interest to the authors, they might consider looking at the Cognitive Bias Modification literature.

We have added the following paragraph to include the value of addressing cognitive biases in an intervention aimed at increasing perception of control:

'In addition, since LoC is the perception of one's control over life events, it is reasonable to consider that such perception can be influenced by biased information processing, that is, by cognitive biases (Jones and Sharpe, 2017). By targeting eventual distorted selectivity in perceiving and elaborating information of one's experiences and preferences, LoC orientation could be steered towards a perception of control.'

Reviewer #2

1. Because the associations are relatively weak (increase in odds for smoking were 14% to 18%, and for hazardous drinking 9% and 1%), these may be particularly vulnerable to unmeasured confounding. However, it's not at all clear what unmeasured factors could confound these associations. From our own

longitudinal research, three possibilities come to mind: childhood behaviour problems (conduct problems; attention problems), exposure to abuse/neglect, and family instability. I don't know whether any of these are significantly associated with the predictor (locus of control), but it seems reasonable to expect that they might be. Do the authors have any data concerning these factors? Are they associated with LoC? If so, it would be helpful to consider these as possible confounders.

We have tested the association between LoC and adverse childhood experiences (using a cumulative measure derived for 0-16 years- see <https://wellcomeopenresearch.org/articles/3-106/v1>) as well as between LoC and conduct disorder (reported by the mother using the strengths and difficulties questionnaire at age 13). We then repeated the analyses including adverse childhood experiences and conduct disorder as confounders. We obtained consistent results with the results reported in the main text (figures 1-2 and SM). In particular, there was evidence that a more external LoC at age 16 was associated with being at least a weekly smoker at age 17 (N = 832; OR 1.13, 95% CI 1.02, 1.25, P = 0.02) and age 21 (N = 819; OR 1.10, 95% CI 1.00, 1.22, P = 0.05). Having a more external LoC at age 16 was also associated with nicotine dependence at age 17 (N = 69; OR 2.03, 95% CI 1.19 to 3.48, P = 0.01), and age 21 (N = 78; OR 1.28, 95% CI 0.84 to 1.95, P = 0.02). A more external LoC at age 16 was associated with higher odds of hazardous drinking on the AUDIT score at age 17 (N = 1105; OR 1.08, 95% CI 1.01 to 1.16, P < 0.03) but not at age 21 (N = 1105; OR 1.00, 95% CI 0.92 to 1.08, P = 0.91). We have not included these analyses in the manuscript because the small sample size available means the estimates are imprecise, but we would be happy to include them if requested.

2. The authors mention that it is not clear what mechanism links LoC to substance use behaviour. One possibility that springs to mind is life stress. It might be possible, for example, to set up a mediation model in which LoC at 16 predicts life stress (if measured, of course) at 17 or 21, to account for a possible mediating pathway. If the authors had such data available, it would add value to this analysis.

We thank the reviewer for a very interesting point. Indeed, a number of studies (for examples: Parkay et al., 1988; Anderson et al., 1977, Bollini et al., 2004), have reported that stress is positively correlated with external LoC and, individuals with more external LoC are more responsive to stress and showed a higher cortisol response. Stress in turn has a complex relationship with nicotine and alcohol consumption; intake of both substances causes stress-like cortisol responses (Lovello 2006).

Subjective reports and biological indices of stress have been implicated in the aetiology of numerous psychological and physical illnesses, including substance addiction. For this reason, we agree that further studies are needed to examine psychological and biological mechanisms that are pivotal in the relationship between LoC, stress and nicotine and alcohol consumption.

We believe that conducting an analysis to include life stress would be an interesting approach, however this is beyond the scope of this manuscript. To our knowledge this current study is the largest and most robust study to date to assess the prospective association between LoC and later assessments of tobacco smoking and alcohol use. We here report, using a large prospective birth cohort which allows for the clear temporal ordering of the variables, further evidence that an external LoC is associated with increased tobacco and alcohol use during adolescence.

Nevertheless, selecting participants from ALSPAC that have a LoC measure, a nicotine and alcohol consumption measure as well as the 'Life events measure' would result in a loss of power by further reducing the sample size. An ad hoc experiment, looking at both the psychological and biological mechanisms that links LoC to stress and then to nicotine and alcohol consumption, would better fit the purpose.

We have added the following sentence in the limitations section:

'Further studies could investigate whether other variables, such as life stress, may account for a possible mediating pathway between LoC and substance use.'

REFERENCES:

Parkay, F. W., Greenwood, G., Olejnik, S., & Proller, N. (1988). A study of the relationships among teacher efficacy, locus of control, and stress. *Journal of Research & Development in Education*, 21(4): 13-22.

Anderson, C. R. (1977). Locus of control, coping behaviors, and performance in a stress setting: A longitudinal study. *Journal of Applied Psychology*, 62(4): 446-451.

Bollini A. M., Walker E. F., Hamann S. E., Kestler L. (2004) The influence of perceived control and locus of control on the cortisol and subjective response to stress. *Biological Psychology*, 67 (3): 245 – 260.

Lovallo W. R. (2006). Cortisol secretion patterns in addiction and addiction risk. *Int. J. Psychophysiol*, 59 (3): 195-202.

Appendix B

Dear Editor,

As suggested we are here including the analyses suggested by R2 (i.e. the ACE confound analyses) in supplementary material.

We have also added the following short paragraph in the main text, at the end of the Statistical Analysis section:

‘Additional analyses, further adjusting for adverse childhood experiences as well as for conduct disorder, are reported in the Supplementary Materials (see Supplementary Material).’

Best Regards,

Glenda Lassi on behalf of all co-authors